# Usefulness of Protocolized Point-of-Care Ultrasonography for Patients with Acute Renal Colic Who Visited Emergency Department: A Randomized Controlled Study

**DOI:** 10.3390/medicina55110717

**Published:** 2019-10-28

**Authors:** Seok Goo Kim, Ik Joon Jo, Taerim Kim, Sung Yeon Hwang, Joo Hyun Park, Tae Gun Shin, Min Seob Sim, Won Chul Cha, Hee Yoon

**Affiliations:** Department of Emergency Medicine, Samsung Medical Center, Sungkyunkwan University School of Medicine, Seoul 06351, Korea; seokgoo1.kim@samsung.com (S.G.K.); taerimi.kim@samsung.com (T.K.); sygood.hwang@samsung.com (S.Y.H.); jhccm.park@samsung.com (J.H.P.); taegun.shin@samsung.com (T.G.S.); wc.cha@samsung.com (W.C.C.)

**Keywords:** ultrasonography, length of stay, emergency, ureterolithiasis, urolithiasis

## Abstract

*Background and Objectives:* Ultrasonography is useful in evaluating patients with renal colic and it has high sensitivity and specificity for diagnosing ureter stones by revealing hydronephrosis. We evaluated the efficacy of point-of-care ultrasonography protocol in managing patients with acute renal colic who visited the emergency department (ED). *Materials and Methods:* Between March 2019 and July 2019, patients who visited the ED because of renal colic were randomly assigned by date of visit either to the conventional group (CG), who underwent routine diagnostic work-up without ultrasonography, or to the ultrasonography group (UG), who underwent bedside ultrasonography as an initial diagnostic testing. When hydronephrosis was detected in the UG group, a confirmatory non-contrast abdomen computed tomography scan was promptly performed. The ED length of stay, complications, and missed or delayed high-risk diagnosis were evaluated. *Results:* In total, 128 of 147 analyzed patients were confirmed to have ureter stones. The ED length of stay was significantly lower in the UG group than in the CG group (mean 172 min; 95% confidence interval (CI): 151–194 min vs. mean 234 min; 95% CI: 216–252 min). The medical cost was also remarkably lower in the UG group than in the CG group (259 USD vs. 319 USD; *p* < 0.001). The incidence of complications within 30 days after visiting ED and missed or delayed high-risk diagnosis were not significantly different between the two groups. *Conclusions:* We found that protocolized point-of-care ultrasonography in patients with acute renal colic who visited the ED can more effectively reduce the length of stay and medical cost without 30-day complication than usual clinical practice.

## 1. Introduction

Urinary colicky pain is a common cause of emergency department (ED) visits. The prevalence of nephrolithiasis and its medical burden have increased in the United States, Europe, Southeast Asia, and South Korea over several decades [1,2,3]. The gold standard imaging modalities for diagnosing urinary stones is non-contrast computed tomography (CT) of the abdomen and pelvis [4,5]. In general, laboratory tests, including urine analysis, are initially performed in patients who are suspected of renal colic in the ED, and a CT scan is done if there is hematuria or no abnormality in other blood tests. However, diagnosis is often delayed because of time-consuming laboratory tests, such as urine analysis in crowded EDs [6]. In addition, the negative predictive value of hematuria for the diagnosis of urinary stone is only 65% [7]. Therefore, ureterolithiasis should not be judged only based on the presence of hematuria, but there are frequent delays in waiting for urinalysis results.

In recent decades, point-of-care ultrasonography (POCUS) has been increasingly used by emergency physicians because of its easy bedside applicability without radiation exposure. Thus, POCUS can be extremely useful for the evaluation of patients with critical conditions visiting the ED [8,9]. Detecting hydronephrosis by ultrasonography is an indirect sign for ureter stone, which has a reliable sensitivity and specificity for the diagnosis of renal colic [4,10,11,12,13,14,15,16]. The importance of POCUS in evaluating patients with renal colic is being increasingly recognized [17,18]. However, only a few ultrasonography algorithms for managing patients suspected with ureter stone exist [19,20,21,22,23].

Therefore, the aim of this study was to describe the POCUS urinary stone (POCUS-US) protocol and to evaluate whether the protocol can effectively reduce diagnostic time for ureter stones or reduce medical expense without complication compared with routine clinical practice in patients with renal colicky pain.

## 2. Materials and Methods

### 2.1. Study Design

This was a prospective randomized controlled trial conducted from March 2019 to July 2019 at the ED of the Samsung Medical Center, an academic tertiary hospital with approximately 78,000 patients annual visit in Seoul, South Korea. The study was approved by the institutional review board committee on January 22, 2019 (IRB File Number: 2018-11-149) and informed consent was obtained from each patient.

### 2.2. Selection of Participants

We enrolled all patients with acute renal colic. However, we excluded patients aged <18 years or patients of any age who were pregnant or had a medical history of progressive intra-abdominal primary or metastatic cancer, urinary stones within the past 30 days, or urogenital abnormality. We also excluded patients with fever (>37.2 °C), trauma patients, patients with a skin lesion with suspected herpes zoster infection at the site of pain, patients who refused to provide informed consent, foreigners who faced difficulty in understanding the purpose of the study because of language differences, and patients who underwent interventional procedures for symptom management or diagnosis in the ED.

### 2.3. Randomization and Study Protocol

Patients who visited the ED because of renal colic were randomly assigned to either the conventional group (CG) or the ultrasonography group (UG) based on the date of visit (Figure 1). Patients who visited the ED on odd days of the month were assigned to CG and underwent routine laboratory blood tests, including complete blood count, electrolytes, chemistry profiles, and urine analyses after history taking and physical examination. Based on test results, an emergency physician decided whether to perform an additional test, such as an abdominal computed tomography (CT) scan with or without contrast.

Patients who visited the ED on even days of the month were assigned to the UG group. These patients underwent bedside ultrasonography as an initial diagnostic testing to evaluate hydronephrosis, simultaneously with physical examination and history taking. If hydronephrosis was detected on bedside ultrasonography, a non-contrast abdominal CT was prescribed as an initial examination according to the protocol. Laboratory tests, which only comprised of urine analysis and creatinine blood test, excluding other routine blood tests, were performed to evaluate obstructive acute kidney injury (AKI) or combined urinary tract infection (UTI). If patients assigned to UG did not have hydronephrosis on bedside ultrasonography, they were examined in the same way as the patients in the CG group (Figure 1).

All patients diagnosed with ureter stone were followed up with 30 days after diagnosis. We checked whether the patients developed a post-renal AKI by Kidney Disease Improving Global Outcomes (KDIGO) staging (Appendix A) or complicated UTI or whether the patient missed or delayed high-risk diagnosis via the electronic medical records (EMR). If the patient did not visit the outpatient urology clinic or did not perform the follow-up laboratory creatinine test within 30 days, the patient was contacted by phone. The following events were queried: Diagnosis of any other disease, UTI-like symptoms, uremic symptoms, ED visit, and follow-up to other hospital (Appendix A).

### 2.4. Ultrasonography

Bedside ultrasonography (US) was performed by 12 attending emergency physicians or 14 residents of the ED who underwent a 4-hour education and 10 cases of practice training. Pre-education consisted of a 1.5-hour didactic session and 2.5-hour hands-on session. The lecture included basic ultrasound principles, anatomy of the kidney and urinary collecting system, and grade of hydronephrosis (Appendix A). The 2.5-hour hands-on session focused on finding the structures of the kidney and urinary system. After the training, they conducted ultrasound examination of 10 cases of sonography to identify the urinary collecting system of kidney and describe the grade of hydronephrosis under the observation of two attending emergency physicians serving as ultrasound instructors.

Bedside US was performed with Samsung ultrasound HM70A (Samsung healthcare, Seoul, Korea). A two-dimensional image mode and color Doppler imaging mode with a 1–7 MHz convex transducer (CA 1–7 AD) on abdominal preset were used for identifying hydronephrosis. We examined both kidneys to make a comparison between normal side and suspected ureter stone side for detecting hydronephrosis with short and long axis views. Additional color Doppler mode was applied to distinguish the renal collection system from the renal vessels. All images were reviewed once a week.

### 2.5. Outcome Measures

The primary outcome was the ED length of stay, and the secondary outcomes were ED medical cost, the incidence of acutely missed or delayed high-risk diagnosis, and complications within 30 days.

### 2.6. Sample Size Calculation and Statistical Analysis

*A priori* sample size calculation was made relative to the primary outcome achievement. We assumed an α value of 0.05 for two-sided hypothesis testing and a β error of 0.20 (power = 80%). We considered that a 30-percentage point reduction of ED length of stay in the UG was clinically significant. We used an independent *t*-test based on the ED length of stay (151 ± 108 min) of the control group who were diagnosed with urinary stone after CT scan during 2017, and approximately 85% of the patients with suspected urinary colic were confirmed. We assumed a drop-out rate of 30%. A total of 152 patients with 76 patients per group was required to detect this hypothesized reduction rate.

Standard descriptive statistics are used to present all data. Continuous variables are given as medians (interquartile ranges (IQRs)) or mean (standard deviation), and the independent *t*-test was used for the mean comparison between the two groups. Categorical data are presented as numbers with percentages and were compared using the chi-square test. STATA ver. 15.0 software (STATA Corporation, College Station, TX, USA) was used to perform all statistical analyses.

## 3. Results

### 3.1. Study Patients and Baseline Characteristics

One hundred and sixty-four patients were enrolled, but 12 patients dropped out because of group mismatching. In total, 152 patients were included in the study, and 76 patients each were randomized to the CG or UG group. During the study period, five patients were excluded in the outcome analysis due to time-consuming interventional procedures in the ED, such as extracorporeal shock wave lithotripsy (ESWL) and double J catheter insertion. Finally, outcomes among 147 patients (74 patients in CG and 73 patients in UG) were analyzed, and of them, 128 were diagnosed with ureter stone (Figure 1). The baseline characteristics of the study participants are shown in Table 1. More than half of the patients in each group were males (57% in CG vs. 67% in UG). The median age was 55 years in the CG group and 48 years in the UG group. However, there were no significant differences in all baseline characteristics, including the numeric rating scale (NRS) and urine analysis results between the CG and UG groups (Table 1).

### 3.2. Outcome Analysis

The ED length of stay was significantly lower in the UG group compared to the CG group. The difference in the length of stay between the CG and UG group was 62 min (UG 172 min (95% CI): 151–194 min] vs. CG 234 min (95% CI 216–252 min), *p*-value for difference <0.001). The subgroup diagnosed as ureter stone in the UG group showed meaningful distinction in terms of length of stay compared with the CG group, 74 min (UG 157 min (95% CI 144–171 min) vs. CG 231 min (211–251 min), *p* -value for difference <0.001 (Figure 2 and Table 2).

Medical cost was significantly lower in the UG than in the CG. The difference in medical cost between the CG and UG at the ED discharge for diagnosing ureter stone was approximately 60 USD (UG 259 (95% CI 240–278) vs. CG 319 (CG 319 USD) USD; *p* < 0.001), and the difference in the subgroup of diagnosed with ureter stone was 63 USD (UG 247 (95% CI 233–260) USD vs. CG 310 (95% CI 298–322) USD; *p* < 0.001) (Figure 3 and Table 2). However, the time from performing CT scan to ED discharge was not significantly different between the CG and UG. Also, there was no significant difference in 30-day complications and missed diagnosis between the CG and UG groups (Table 2).

## 4. Discussion

In this study, we found that patients with hydronephrosis on initial bedside ultrasonography had shorter ED length of stay than the CG group. Thus, ED length of stay could be reduced by 32% without serious 30-day complications and delayed high-risk diagnosis compared to the CG. The medical cost for the UG with hydronephrosis in the ED was approximately 20% lower compared to the CG. Therefore, the application of initial ultrasonography to patients with acute renal colic proved to be a safe and efficient operation within the ED.

In several studies, the sensitivity and specificity of ultrasonography in the detection of hydronephrosis by emergency physician were 72–83% and 83%, respectively [14,15,24]. In case of moderate or greater hydronephrosis, the specificity was 94.4% [16]. In this study, 72% sensitivity and 83% specificity were shown compared to CT, which are similar to findings in previous studies. Ultrasonography is an operator dependent technique. Furthermore, 11–15% of patients with ureter stone may not have hydronephrosis which is associated with the patient’s hydration state [15,19,25,26]. When applying ultrasonography in this study, despite the presence of hydronephrosis in 67% of patients (Appendix A), ED length of stay significantly decreased in the overall UG. This represents a good example of a protocolized application of ultrasonography, which was able to effectively guide the treatment and disposition of the patient.

The length of stay in the ED from CT performing time to discharge time was not significantly different between the CG and UG (CG 104 min (CI: 91–118 min) vs. UG 125 min (CI: 104–141min); *p* < 0.07). This shows that the final diagnosis and the patient’s disposition depend on when CT was performed in patients with renal colic. UG can shorten the CT performing time based on bedside ultrasonography results of hydronephrosis presence without awaiting other test results. Therefore, applying POCUS-US protocol for diagnosing ureter stone in patients with renal colic at initial approach can reduce the CT performing time and accordingly shorten the diagnosis time and length of stay for ureter stones in the ED.

The incidence of complications within 30 days and missed or delayed high-risk diagnosis were not significantly different between the two groups, and there was not any serious complication and delayed high-risk diagnosis reported. This is owing to a protocol-based diagnostic process, and abdominal non-contrast CT can also help diagnose causes other than ureter stones [5,27]. One patient was a 33-year-old male who presented with left flank pain without any previous medical history. Evaluated as a mild hydronephrosis on the initial bedside ultrasonography, the abdominal non-contrast CT was performed, but there was no hydronephrosis and ureter stones on CT. The patient complained of persistent progressive abdominal pain and we promptly performed further examination according to the protocol. There was an aneurysm of splenic artery with rupture sign, so the patient underwent a coil embolization procedure at the bleeding artery. In this case, initial subtle hydronephrosis could be misinterpreted, but the abdominal non-contrast CT complementally revealed the negative sign of hydronephrosis and ureter stone. We considered an additional contrast CT clinically. Thus, we did not miss the serious and high-risk diagnosis.

Previous algorithms for the diagnosis of ureter stone were based on risk stratification with clinical symptoms and urine analysis results to inform the decision to perform abdomen CT [5,6,19,27]. However, the gold standard for diagnosis of ureter stone is abdominal non-contrast CT, with sensitivity and specificity of 95–98% [28]. It is important to minimize fatal misdiagnosis because our institution is the tertiary general hospital, and most patients suspected of having ureter stone due to high medical coverage in South Korea undergo CT scans. Therefore, after excluding probability for other diseases, such as intraperitoneal cancer, previous renourinary diseases, and fever in patients who came to the ED for renal colic, we performed the CT scan when the hydronephrosis was detected on the ultrasonography at the initial approach without waiting for unnecessary laboratory tests.

The most important thing for patients in the ED is safety and efficiency of ED management. ED overcrowding is widespread and has reportedly reached crisis proportions in many countries [29,30]. Many hospitals are struggling to reduce ED crowding and increase patient safety through multimodal interventions on patient flow in the ED [31,32]. If we use bedside ultrasonography to identify hydronephrosis in patients with acute renal colic and then perform non-contrast abdomen CT promptly if the hydronephrosis exist at initial approach, we can shorten the CT performing time to diagnose for ureter stones without fatal misdiagnosis and can significantly reduce length of stay and medical expense in the ED.

### Limitations

There are several limitations. First, this study was randomized by date of visit, but physicians were not blinded. Thus, there could be possibility of bias. Second, patients often experienced longer length of stay due to overcrowding in the ED. However, we have not considered other overcrowding factors that delay the ED length of stay. Third, we performed abdomen CT for all enrolled patients with acute renal colic without risk stratification. Fourth, while some patients might have stayed longer to facilitate pain control, we did not consider the interrelation between pain control time and ED length of stay. Fifth, for patients who did not complete follow-up, the telephone interview regarding uremic symptoms could not substitute the actual kidney function for measured value by laboratory blood test. Thus, it is possible that we underestimated the results of 30-day complications. Sixth, since ultrasound is operator dependent, the application may vary depending on the hospital and the environment. Seventh, as a result of the exclusion criteria, we enrolled clinically well patients with non-urgent renal colic. Thus results could not be generalizable for clinically complicated patients. Last, this study was performed at a single center and therefore the results may not be generalizable to other settings. Hence, further studies in other settings are needed.

## 5. Conclusions

Applying the initial diagnostic POCUS-US protocol for patients with acute renal colic in the ED can reduce the ED length of stay and medical expense without serious complications.

## Figures and Tables

**Figure 1 medicina-55-00717-f001:**
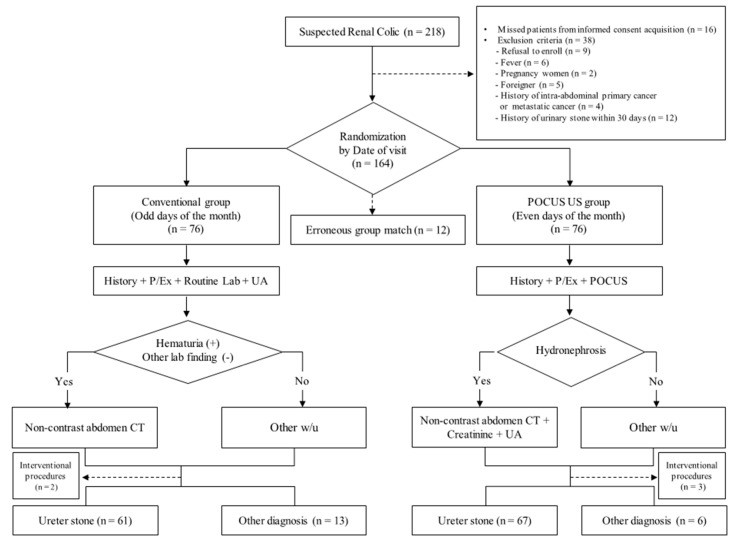
Study flow and point-of-care ultrasonography urinary stone (POCUS-US) protocol. P/Ex, physical examination; lab, laboratory blood test; UA, urine analysis; CT, computed tomography; w/u, work up; MRCP, magnetic resonance cholangiopancreatography; ERCP, endoscopic retrograde cholangiopancreatography; ESWL, extracorporeal shock wave lithotripsy.

**Figure 2 medicina-55-00717-f002:**
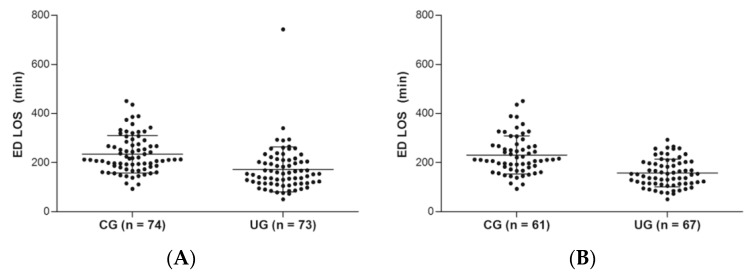
Length of stay comparison by scatterplot between conventional group and ultrasonography group: (**A**) All patients including other diagnosis; (**B**) the patients diagnosed as ureter stone. ED, emergency department; LOS, length of stay; min, minutes.

**Figure 3 medicina-55-00717-f003:**
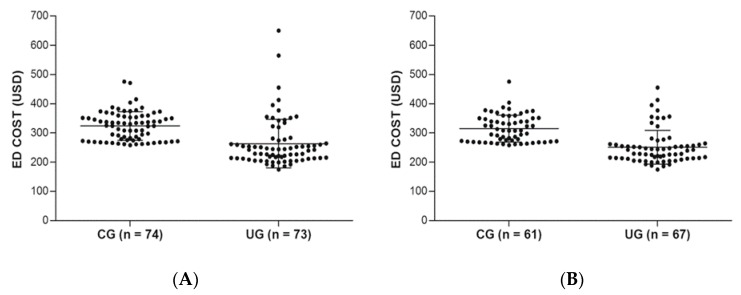
Medical expense comparison by scatterplot between the conventional group and ultrasonography group at the ED discharge: (**A**) All patients, including other diagnosis; (**B**) The patients diagnosed with ureter stone. ED, emergency department; USD, united states dollar(s).

**Table 1 medicina-55-00717-t001:** Baseline characteristics of study patients.

	Conventional Group (n = 74)	Ultrasound Group (n = 73)	*p*-Value
Male	42 (57)	49 (67)	0.196
Age (years)	55 (42–62)	48 (38–59)	0.164
Previous medical history			
Hypertension	9 (12)	10 (14)	0.781
Diabetes	1 (2)	4 (6)	0.209
Cardiac, Lung, Kidney disease	0	0	0.999
Malignancy	2 (3)	3 (4)	0.681
Liver disease	1 (2)	3 (4)	0.304
Ureter stone history (before 30 days)	18 (24)	22 (30)	0.429
Pain score (Numeric rating scale)	6 (4–7)	5 (4–7)	0.584
Vital sign			
SBP (mmHg)	139 (120–157)	134 (120–155)	0.173
DBP (mmHg)	83 (74–97)	83 (75–93)	0.651
Heart rate (beats per minute)	73 (66–81)	74 (66–79)	0.744
Body temperature (°C)	36.5 (36.3–36.9)	36.6 (36.2–36.8)	0.889
Laboratory test			
Serum creatinine	0.93 (0.78–1.02)	0.97 (0.76–1.49)	0.258
Urine analysis			
Presence of Hematuria	56 (76)	63 (86)	0.101
Nitrite	3 (4)	3 (4)	0.999
Leukocyte esterase	15 (20)	16 (22)	0.807
Admission	0	0	0.999
Final diagnosis of ureter stone	61 (83)	67 (92)	0.139

Data are shown as median (interquartile range) or number (%). SBP, systolic blood pressure; DBP, diastolic blood pressure.

**Table 2 medicina-55-00717-t002:** Outcome analysis of study patients.

	Conventional Group (n = 74)		Ultrasound Group (n = 73)		*p*-Value
	Mean (±SD)	95% CI	Mean (±SD)	95% CI	
ED LOS (min)	234 (±76)	(216–252)	172 (±92)	(151–194)	<0.001
Time from CT to discharge (min)	104 (±60)	(91–118)	125 (±70)	(104–141)	0.065
ED Cost (USD)	319 (±48)	(308–330)	259 (±81)	(240–278)	<0.001
Diagnosed as Ureter stone	CG (n = 61)		UG (n = 67)		
ED LOS (min)	231 (±78)	(211–251)	157 (±57)	(144–171)	<0.001
Time from CT to discharge (min)	104 (±62)	(88–119)	118 (±50)	(106–130)	0.154
ED Cost (USD)	310 (±46)	(298–322)	247 (±57)	(233–260)	<0.001
Follow-up					
ESWL	14 (23)		14 (21)		0.813
AKI	0 (0)		1 (2)		0.999
UTI	0 (0)		1 (2)		0.999
Other hospital follow-up	6 (10)		7 (11)		0.886
Missed diagnosis	0		0		

Data are shown as mean (± standard deviation) with 95% confidence interval or number (%). SD, standard deviation; CI, confidence interval; CG, conventional group; UG, ultrasound group; ED, emergency department; LOS, length of stay; CT, computed tomography; min, minute; USD, United States dollar(s); ESWL, extracorporeal shock wave lithotripsy; AKI, acute kidney injury; UTI, urinary tract infection.

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
