# Peer review of "Usefulness of Protocolized Point-of-Care Ultrasonography for Patients with Acute Renal Colic Who Visited Emergency Department: A Randomized Controlled Study"

_medicina, 2019, doi:10.3390/medicina55110717_

Round 1

Reviewer 1 Report

This is an interesting study demonstrating the importance of immediate ultrasound examination in patients with flank pain / renal colic. The value of ultrasound as an initial examination has been addressed already in 2005 by Strohmaier: Diagnostic approach. In: Ch. Chaussy et al.: Therapeutic energy applications in urology. Thieme Verlag stuttgart New York, pp. 65-71. This should be referenced.

Only some minor issues should be addressed:

I do not understand why waiting for urinalysis is a frequent delay. Urinalysis can be done as a point-of-care method using a dipstick. So the result is available immediately. The combination of microhaematuria and hydronephrosis on ultrasound will make the diagnosis of urolithiasis even more probable.

The statistical evaluation is potentially inappropriate. Student´s t-test may be used only in case of equal variance and Gaussian distribution. this has to be checked by appropriate tests first.

Author Response

Response to Reviewer 1 Comment

Comments and Suggestions for Authors

This is an interesting study demonstrating the importance of immediate ultrasound examination in patients with flank pain / renal colic.

Point 1: The value of ultrasound as an initial examination has been addressed already in 2005 by Strohmaier: Diagnostic approach. In: Ch. Chaussy et al.: Therapeutic energy applications in urology. Thieme Verlag stuttgart New York, pp. 65-71. This should be referenced.

Response 1: We appreciate the reviewer’s helpful comments. We have cited this reference in the revised manuscript as per your advice on P2 line 48-49.

Only some minor issues should be addressed:

Point 2: I do not understand why waiting for urinalysis is a frequent delay. Urinalysis can be done as a point-of-care method using a dipstick. So the result is available immediately. The combination of microhaematuria and hydronephrosis on ultrasound will make the diagnosis of urolithiasis even more probable.

Response 2: Routine urine analysis (UA) in our institution includes microscopic analysis and dipstick test; thus the turnaround time is often more than 1.5–2 hours. In addition, our institution is a tertiary university hospital with more than 200 emergency department visitors every 24 hours ; therefore is heavily overcrowded. Moreover, some patients may not be able to urinate easily because of urinary tract obstructions caused by ureterolithiasis. Therefore, the UA test can lead to delays in diagnosis. We designed this study to provide evidence for the usefulness of performing a CT scan using ultrasound without waiting for the urine test results to diagnose patients with suspected renal colic.

Point 3: The statistical evaluation is potentially inappropriate. Student´s t-test may be used only in case of equal variance and Gaussian distribution. this has to be checked by appropriate tests first.

Response 3: We performed the ‘Shapiro-Wilk test’, which fits the standard normal distribution. However, the variances differed in the ED length of stay between the two groups diagnosed as ureter stone and the cost between groups. Therefore, we used a two-sample t test with unequal variances for these analyses.

Reviewer 2 Report

Review

Usefulness of protocolized point-of-care ultrasonography for patients with acute renal colic who visited emergency department: a randomized controlled study

The authors prospectively tried to evaluate ultrasonography of the kidneys in an emergency setting for suspected renal colic.

Abstract

Ok

Keywords

Ok

Introduction

P2line44: I strongly doubt that POCUS is just recently increasingly used by physicians at the ED. Actually it is part of all accepted guidelines in the emergeny setting. Please adjust or refer to guidelines.

P2line47: US does not diagnose renal colic, but may diagnose the responsible stone. Please adjust. Have a look at this paper (PMID 27459091). It strongly contradicts your statement that US has a high sensitivity for the diagnosis of renal stones. If you go for hydronephrosis only you may be right. Please comment and specify.

P2line50: ref.16-18: please include established guidelines.

Materials and methods

Ok

Results

P4line 132: I do not understand the numbers given. You enrolled 164 patients. 28 dropped out, so you included 152 (?). These numbers do not math with fig1 either. Did the exclusion of the 5 time consuming patients might have affected your outcome?

P5line152: If you adjust fig2 including the 5 excluded time consuming patients, how does it change the results?

P6line163: how long did it take to perform US? Please specify. Refering to recent guidelines it is no mistake to directly perform non-enhanced low dose CT for workup of suspected renal colic. This pathway is much less time consuming than the ones described. Please comment. Why did you not include it in your analysis?

P6line163: Honestly I’m not sure if this very compact overview of the estimated costs allows you to draw conclusions. Did you consider ALL costs, what about costs of complications, false negative or false positive results. I think your study design is not made to give answer. Please comment.

Discussion

P6line182: please give a metaanalysis as reference for diagnostic accuracy of US.

P6line178: again, I think your estimation of costs is a really very rough one.

P6line183: If I understood right, not all patients underwent CT. How will you calculate your sensitivity/ specificity if you do not have alle positives/ negatives? You can estimate you test accuracy for the subgroup that underwent CT. what is the gold standard for hydronephrosis?

Author Response

Response to Reviewer 2 Comment

Comments and Suggestions for Authors

Review

Usefulness of protocolized point-of-care ultrasonography for patients with acute renal colic who visited emergency department: a randomized controlled study

The authors prospectively tried to evaluate ultrasonography of the kidneys in an emergency setting for suspected renal colic.

Abstract

Ok

Keywords

Ok

Introduction

Point 1: P2line44: I strongly doubt that POCUS is just recently increasingly used by physicians at the ED. Actually it is part of all accepted guidelines in the emergeny setting. Please adjust or refer to guidelines.

Response 1: I agree with your comments. As mentioned, POCUS is becoming an increasingly prevalent diagnostic tool in the ED as it can be used for diagnosis of many emergency conditions. We have revised the sentence as follows:

(P2, line 44-47)

“In recent decades, point-of-care ultrasonography (POCUS) has been increasingly used by emergency physicians because of its easy bedside applicability without radiation exposure; thus POCUS can be extremely useful for the evaluation of patients with critical conditions visiting the ED.”

Point 2: P2line47: US does not diagnose renal colic, but may diagnose the responsible stone. Please adjust. Have a look at this paper (PMID 27459091). It strongly contradicts your statement that US has a high sensitivity for the diagnosis of renal stones. If you go for hydronephrosis only you may be right. Please comment and specify.

Response 2: PMID 27459091 is a retrospective study showing that the diagnosis of renal stones with ultrasound is less sensitive, and the diagnosis is related to the size of the stone. Historically, renal US has been reported to have fair sensitivity (37%–64%) for detecting renal stones; however, renal US is more sensitive for identifying acute renal obstruction, like hydronephrosis.

In the presence of hydronephrosis, the sensitivity and specificity of US were 81% and 97.5%, respectively, for ureterolithiasis (PMID: 25168527) and in case of moderate or greater hydronephrosis presence, the specificity was 94.4% (PMID: 29427476).

Therefore, we aimed to assess hydronephrosis, not for renal stones using ultrasound in this study. We have provided the following information in the text:

(P2 lines 47-48)

Detection of hydronephrosis by ultrasonography is an indirect indication of ureter stones (PMID: 24578772, PMID: 9848702, and PMID: 16282510). The technique has reliable sensitivity and specificity for the diagnosis of ureterolithiasis in patients with renal colicky pain.

Point 3: P2line50: ref.16-18: please include established guidelines.

Response 3: We have cited the appropriate references in the revised manuscript, namely, PMID: 12074804 and PMID: 26616536.

Materials and methods

Ok

Results

Point 4: P4line 132: I do not understand the numbers given. You enrolled 164 patients. 28 dropped out, so you included 152 (?). These numbers do not math with fig1 either.

Response 4: We apologize for the error. Twelve patients dropped out due to group mismatching. We have revised the manuscript and provided the correct data, as follows:

(P4 lines 134–135)

One hundred and sixty-four patients were enrolled, but 12 patients dropped out because of group mismatching”

Point 5: Did the exclusion of the 5 time consuming patients might have affected your outcome?

Response 5:  Patients who underwent interventional procedures in the ED were strongly suspected of having different clinical diseases, such as ovarian torsion, CBD stone, and splenic artery occlusion, which resulted in longer ED length of stay during screening. In addition, these diseases required immediate treatment, such as ERCP, and embolization, together with the diagnosis of the condition; therefore, the ED length of stay of such patients was much longer. As this study used the ED length of stay as the primary outcome, the patients who underwent these procedures in the ED could introduce bias in the results. Therefore, we formulated the exclusion criteria accordingly, so that these patients were excluded from the study.

Comparisons between groups including patients diagnosed as having other diseases were affected. However, the comparisons between the groups diagnosed as having ureteral stones did not change, including the groups with patients who underwent ESWL or D-J insertion in the ED.

Point 6: P5 line 152: If you adjust fig2 including the 5 excluded time consuming patients, how does it change the results?

Response 6: Comparisons between groups including patients diagnosed with other diseases were mildly affected. However, the comparisons between the groups diagnosed with ureteral stones did not change, including the groups with patients who underwent ESWL or D-J insertion in the ED. As mentioned in A5, this could introduce a bias in the results; hence, we excluded these patients according to the enrollment criteria. We believe that this has no significant effect on the intent and conclusion of our study.

Point 7: P6 line 163: how long did it take to perform US? Please specify. Referring to recent guidelines it is no mistake to directly perform non-enhanced low dose CT for workup of suspected renal colic. This pathway is much less time consuming than the ones described. Please comment. Why did you not include it in your analysis?

Response 7: We could not measure the time taken in performing US directly. However, an ultrasound machine was placed in the examination room of our ED, and the researchers performed ultrasound examinations on the patients simultaneously, with physical examination and history taking. In addition, we performed a simply protocolized ultrasound scan to detect only hydronephrosis (short and long axis views of both kidneys and color Doppler). First, we selected the ED length of stay as the primary outcome. The ED length of stay is the total ED stay time from the time of visit of the patient to the time of their discharge. Therefore, time taken for performing the ultrasound was already included in the ED length of stay and did not affect the results.

As you mentioned, non-contrast low-dose CT of the abdomen is currently thought to be the preferred imaging study for most patients (PMID: 26318710). Therefore, we aimed to quickly determine if a CT scan is needed for patients with suspected renal colic by performing ultrasound examination at the initial prescription stage. This not only reduces the ER length of stay, but also avoids unnecessary laboratory testing, which is important for efficient ED operation.

Point 8: P6line163: Honestly I’m not sure if this very compact overview of the estimated costs allows you to draw conclusions. Did you consider ALL costs, what about costs of complications, false negative or false positive results. I think your study design is not made to give answer. Please comment.

Response 8: We calculated all costs, including those incurred for false-negative and false-positive results. Instead, we only examined the medical costs actually incurred at the ED. South Korea has a unique medical insurance system as the national health insurance. Approximately 97% of the population (~52 million persons) is included in the system (PM ID: 27794523). This is a mandatory system, and medical costs are very low. Thus, medical costs in addition to those assessed by us are rarely incurred. Although our study design is not perfect for evaluation of the cost-effectiveness, we believe that our ED costs analysis is reliable.

Discussion

Point 9: P6line182: please give a metaanalysis as reference for diagnostic accuracy of US.

Response 9: In a meta-analysis study conducted by the University of Calgary, Wong C et al., examined the accuracy and prognostic value of point of care ultrasound for nephrolithiasis in the ED. Their study showed that point of care ultrasound has modest diagnostic accuracy for diagnosing nephrolithiasis (PMID: 29427476).

We have cited the reference in the revised manuscript, as per your advice, and have added the flowing sentence on P6, lines 184-185.

In case of moderate or greater hydronephrosis, the specificity was 94.4%”

Point 10: P6 line 178: again, I think your estimation of costs is a really very rough one.

Response 10: We understand your concern. However, we calculated all costs incurred during the ED stay. South Korea has a unique medical insurance system as the national health insurance. This is a mandatory system, and medical costs are very low. Thus, medical costs in addition to those assessed by us are rarely incurred. Although our study design is not perfect for the evaluation of cost-effectiveness, we believe that our ED costs analysis is reliable.

Point 11: P6line183: If I understood right, not all patients underwent CT. How will you calculate your sensitivity/ specificity if you do not have all positives/ negatives? You can estimate you test accuracy for the subgroup that underwent CT. what is the gold standard for hydronephrosis?

Response 11: In this study, all enrolled patients underwent a CT scan with or without contrast enhancement, based on the results of the laboratory tests or ultrasound. Thus, the sensitivity and specificity of ultrasound could be calculated in comparison with those of CT for the detection of hydronephrosis. The accuracy of ultrasound for detecting ureterolithiasis was 73% in this study.

Based on the current guidelines, the gold standard imaging tool for ureterolithiasis is abdominal non-contrast CT (PMID: 14960744 and PMID: 26318710). However, ultrasound also has reliable sensitivity (72–83%) and specificity (83%) compared with CT (PMID: 16282510, PMID: 17564683 and PMID: 9848702) for the detection hydronephrosis. In addition, as a radiation hazard is not associated with ultrasound, it is considered useful in the initial diagnostic planning for patients with renal colic and their follow-up.

Reviewer 3 Report

Review of original paper by Seok Goo Kim et al. with the title “Usefulness of protocolized POCUS for patients with acute renal colic who visited emergency department: a randomized controlled study".

The manuscript needs grammatical proof-reading before being relevant for publication (i.e.: Page 1, line 40: Please consider rephrase the sentence and dividing sentence into two. Page 2, line 63: This sentence is far too long. Consider shorter sentences. Line 209: "contrast" instead of "contract").

Please explain in detail how the different US measures of hydronephrosis are used currently in clinical practice.

I work in a tertiary hospital, where CT is 24 hours available, even that, abdomen CT is not usually performed in uncomplicated renal colic. I am curious about the number of abdominal CT performed in this study, might seem to many.

In table 1 you show the baseline characteristics of study patients. In the UG group, routine labs and urine analysis were ordered as well? Ureter stone hystory (within 30 days)/malignancy were not an exclusion criteria?

In Table 2, could you add the variable TIME of "arrival to the ED to first physician contact" and "physician contact to blood extraction/urine collection or POCUS exam"? If not recorded, that should be listed as a limitation that could bias the results.

The incidence of complications at follow-up seems really low. Did you only recruit "uncomplicated" renal colic? With 0 admissions in each group, is hard to believe.

Could you explain further the reason why the 28 patients dropped out?

Although I consider myself a POCUS fan, I am reluctant to think that one can rule out hydronefrosis based on a 4 hour training. Did they have any US experience prior this study? How long did it take to finish a whole study? Just like the given example, from the UG, how many false negative and false positive studies did you find?

Due to the questions arised, my impression is that the generalisability and applicability may be limited.

Author Response

Response to Reviewer 3 Comment

Comments and Suggestions for Authors

Review of original paper by Seok Goo Kim et al. with the title “Usefulness of protocolized POCUS for patients with acute renal colic who visited emergency department: a randomized controlled study".

Point 1: The manuscript needs grammatical proof-reading before being relevant for publication (i.e.: Page 1, line 40: Please consider rephrase the sentence and dividing sentence into two. Page 2, line 63: This sentence is far too long. Consider shorter sentences. Line 209: "contrast" instead of "contract").

Response 1: We thank you for the helpful advice. We have rephrased the sentence in the revised manuscript, as per your advice (P1 line 40–P2 line 43).

“In addition, the negative predictive value of hematuria for the diagnosis of urinary stone is only 65%. Therefore, ureterolithiasis should not be judged only based on the presence of hematuria, but there are frequent delays in waiting for urinalysis results.”

We have made the following revisions in the revised manuscript, as per your advice:

(P2 lines 63–70)

“We enrolled all patients with acute renal colic. However, we excluded patients aged <18 years or patients of any age who were pregnant or had a medical history of progressive intra-abdominal primary or metastatic cancer, urinary stones within the past 30 days, or urogenital abnormality. We also excluded patients with fever (>37.2°C), trauma patients, patients with a skin lesion with suspected herpes zoster infection at the site of pain, patients who refused to provide informed consent, foreigners who faced difficulty in understanding the purpose of the study because of language differences, and patients who underwent interventional procedures for symptom management or diagnosis in the ED.”

We have rephrased the following sentence in the revised manuscript, as per your advice:

(P7 line 212-213)

We considered performing additional contrast CT scans clinically; thus, we did not miss the serious and high-risk diagnosis.”

Point 2: Please explain in detail how the different US measures of hydronephrosis are used currently in clinical practice.

Response 2: We performed a simple protocolized ultrasound scan to detect hydrocephalus (short and long axis views of both kidneys and color Doppler). We described the grade of hydronephrosis in Supplement 3.

Supplement 3. Grade of hydronephrosis

Hydronephrosis grade (Point of Care Ultrasound, 1e)

Mild

Enlargement of the renal calices with preservation of the renal papillae

Moderate

Rounding of the renal calices, obliteration of renal papillae, and blunting of medullary pyramids

severe

Calyceal ballooning, complete obliteration of papillae and pyramids, and cortical thinning

Point 3: Line 209: "contrast" instead of "contract").

Response 3: We rephrased the following sentence in the revised manuscript, as per your advice:

(P7 line 212-213)

We considered performing additional contrast CT scans clinically; thus we did not miss the serious and high-risk diagnosis.”

Point 4: I work in a tertiary hospital, where CT is 24 hours available, even that, abdomen CT is not usually performed in uncomplicated renal colic. I am curious about the number of abdominal CT performed in this study, might seem to many.

Response 4: In this study, all enrolled patients underwent a CT scan with or without contrast based on the laboratory tests results or ultrasound. Non-contrast low-dose CT of the abdomen is currently considered the preferred imaging study for most patients (PMID: 26318710). We designed this study based on the concept, “Detection of hydronephrosis by ultrasonography is an indirect indication of ureter stone” to provide evidence for the usefulness of performing a CT scan without awaiting other test results for diagnosing ureterolithiasis in patients with acute renal colic. Moreover, in South Korea, most patients suspected of having a ureter stone undergo CT scans, given the high medical insurance coverage.

Although CT scans have superior diagnostic characteristics, patients are also exposed to greater amounts of ionizing radiation during CT. Therefore, we mentioned this point as a limitation of the study, as follows

(P7, line 235-236)

 “Third, we performed abdominal CT for all enrolled patients with acute renal colic without risk stratification.”

Point 5: In table 1 you show the baseline characteristics of study patients. In the UG group, routine labs and urine analysis were ordered as well?

Response 5: For patients with hydronephrosis in the UG group, as described on P3 line 87–89, only urine analysis and creatinine blood test were performed to evaluate obstructive acute kidney injury (AKI) or combined urinary tract infection (UTI).

Point 6: Ureter stone hystory (within 30 days)/malignancy were not an exclusion criteria?

Response 6: We have rephrased the following text in the revised manuscript, as per your advice:

 (Table 1)

“Ureter stone history (before 30 days)”

Our exclusion criteria for malignancy was only “medical history of progressive intra-abdominal primary or metastatic cancer”. Patients with malignancies, as listed in Table 1, had a medical history of other kinds of tumors, such as breast cancer and thyroid cancer without tumor recurrence and metastasis in the abdomen.

Point 7: In Table 2, could you add the variable TIME of "arrival to the ED to first physician contact" and "physician contact to blood extraction/urine collection or POCUS exam"? If not recorded, that should be listed as a limitation that could bias the results.

Response 7: We did not record these times. The variation in these durations may be large due to overcrowding in the ED; therefore we did not check them because it was considered to be an inaccurate parameter. In addition, we conducted a randomized trial to minimize these effects and applied equally to both groups.

First, we planned the ED length of stay as the primary outcome, with ED length being the total ED stay time from the time of visit to the time of discharge of the patients. Therefore, performing taken for performing the ultrasound was already included in the ED length of stay and did not affect the results.

However, we did not consider these factors that increase the ED length of stay; therefore, we have describe this as a limitation, as follows:

(P7 lines 234–236)

“Second, patients often experienced longer length of stay due to overcrowding in the ED; however, we have not considered other overcrowding factors that delay the ED length of stay.”

Point 8: The incidence of complications at follow-up seems really low. Did you only recruit "uncomplicated" renal colic? With 0 admissions in each group, is hard to believe.

Response 8: As described on P2 line 66, we excluded patients who had fever (>37.2°C) at the time of visitation. If the urine analysis test results were positive for urine nitrate, urine leukocyte esterase and more than 4–5 leukocytes were detected in the field of vision during microscopic examination of the urine, we prescribed empirical antibiotics to the patient. This may have led to fewer complications, such as UTI.

As described in Supplement 4., 33% of patients exhibited what looked like normal collecting duct without hydronephrosis and 60% of patients had mild hydronephrosis. Therefore, there may be fewer obstructive AKI requiring admission. In addition, our hospital is a tertiary university hospital with a high proportion of patients with severe conditions and a large number of patients awaiting hospitalization for treatment. Therefore, procedures that could be carried out in the ED were done in the ED and the patients were discharged without admission after the procedure. Since this may be a limitation of a single center study, we described it as a limitation as follows:

(P8, line 244-245)

“Last, this study was performed at a single center and therefore the results may not be generalizable to other settings. Hence, further studies in other settings are needed.”

Point 9: Could you explain further the reason why the 28 patients dropped out?

Response 9: We apologize for the errors in reporting this number. Twelve patients dropped out due to group mismatching. We have revised the manuscript accordingly, on P4 lines 134–135.

“One hundred and sixty-four patients were enrolled, but 12 patients dropped out because of group mismatching.”

Point 10: Although I consider myself a POCUS fan, I am reluctant to think that one can rule out hydronefrosis based on a 4 hour training. Did they have any US experience prior this study?

Response 10: They received 4 h of ultrasound training for the detection of hydronephrosis. After the training, they conducted ultrasound examination of 10 cases of sonography to identify the kidney’s urinary collecting system and describe the grade of hydronephrosis under the observation of two attending emergency physicians serving as ultrasound instructors.

As described in Supplement 4, all of the ultrasound practitioners were at least two years of emergency residents or emergency professionals. They all had at least 30 cases of previous renal ultrasound experiences, of which more than 50% stated that they had more than 50 cases of renal ultrasound experiences before. Therefore, we thought they had no difficulty to detect hydronephrosis.

Point 11: How long did it take to finish a whole study?

Response 11: This study was carried out from March 2019 to July 2019. Our institution is a tertiary university hospital with more than 200 ED visitors every 24 hours, and with one or two patients being enrolled per day.

Point 12: Just like the given example, from the UG, how many false negative and false positive studies did you find?

Response 12: As we described in P6 lines 185-186, the sensitivity and specificity of ultrasonography in the detection of hydronephrosis were 72% sensitivity and 83%, respectively compared to CT in this study. There was only one false positive case among 49 cases, and there were 19 false negative cases among 24 cases. As shown in supplement 4, because 33% of patients looked like normal collecting duct without hydronephrosis and 60% of patients had mild hydronephrosis, there seemed to be a lot of false negatives. Since ultrasound is operator dependent, we described this as a limitation, as follows:

(P7, line 240-241, P8, line 242)

“Sixth, since ultrasound is operator-dependent, the application may vary depending on the hospital and the environment.”

Reviewer 4 Report

Authors performed a trial of POCUS in the ED setting in patients presenting with renal colic, randomized to usual ED evaluation including labs and CT or expedited POCUS with subsequent CT is hydronephrosis was detected. Primary outcome was ED LOS; other outcomes were cost of stay and missed diagnoses. Authors found that patients in the POCUS group had shorter LOS and lower cost without significantly higher missed diagnoses. Authors conclude that a POCUS protocol in the ED for renal colic can reduce LOS and cost.

Was the trial protocol prospectively registered in a clinical trial registry? If so, please include registration details; if not, explain why. Line 85, scan was "promptly" performed. Does this mean that pts were expedited to obtain CT? This would be a factor in total LOS. What were the factors/criteria for discharge?  Were urology consultants involved for any pts who were diagnosed with stones? If yes, in some or all pts? Involvement of a consultant would be a factor in LOS. Lines 132-133, please correlate with group numbers (N) in Fig 1. It appears that 164 pts were enrolled and randomized, and 12 were removed due to group match erros, but where are the 28 pts who dropped out? If "dropped out" means excluded, this should not be mentioned as part of the enrolled sample. Please describe recruitment and enrollment procedures in more detail. Did enrollment in both groups continue until each group reached exactly 76 patients? If so, did both groups reach the target at the same time? When and how were enrollment concluded? Table 1, please include comparative statistics between the groups. Authors should make clear that, based on their a priori exclusion of patients with complex history or with diagnoses requiring immediate intervention, the context of this trial is limited to clinically well patients with non-urgent renal colic and/or ureteral stones. Authors correctly point out that the timed saved in the LOS between the groups appears to have been from ED presentation to CT. It appears from the protocol that ED staff waited until results of all routine labs and UA were available to determine whether pts in the conventional group should then have CT. In routine practice, however, it would seem that the decision to pursue CT would be made in parallel rather than serially. Ultimately, if clinical suspicion (in addition to labs, UA, and/or POCUS) warrants cross-sectional imaging, it would be expected that ED staff would obtain CT, thus shortening the time to CT. Line 216, "are undergo CT scans" check grammar There is some evidence that clinically stable patients with high suspicion of stones on US may not need CT and can thus avoid radiation exposure. Do the findings of the present study have implications for US-only evaluation in the ED?

Author Response

Response to Reviewer 4 Comment

Comments and Suggestions for Authors

Authors performed a trial of POCUS in the ED setting in patients presenting with renal colic, randomized to usual ED evaluation including labs and CT or expedited POCUS with subsequent CT is hydronephrosis was detected. Primary outcome was ED LOS; other outcomes were cost of stay and missed diagnoses. Authors found that patients in the POCUS group had shorter LOS and lower cost without significantly higher missed diagnoses. Authors conclude that a POCUS protocol in the ED for renal colic can reduce LOS and cost.

Point 1: Was the trial protocol prospectively registered in a clinical trial registry? If so, please include registration details; if not, explain why.

Response 1: The diagnosis process in both groups somewhat changed depending on the physician in charge, but most diagnosis processes are routinely carried out in our hospital. Therefore, since no unnecessary or unusual interventions were actually applied during the study phase, we conducted the study under IRB approval without registration in the clinical trial. But, we believe that we should have registered in the clinical trial registry, and we will register in the next prospective study.

Point 2: Line 85, scan was "promptly" performed. Does this mean that pts were expedited to obtain CT? This would be a factor in total LOS.

Response 2: As you mentioned, the word "promptly" may be confusing. We meant that, if hydronephrosis was detected on bedside ultrasonography in the UG, non-contrast abdominal CT was prescribed at the same time of initial prescription according to the protocol. Therefore, we have rephrased the following sentence in the revised manuscript:

(P3 lines 85–87)

“If hydronephrosis was detected on bedside ultrasonography, non-contrast abdominal CT was prescribed as an initial examination according to the protocol”.

However, because physicians were not blinded, we have mentioned this as a limitation of the study, as follows:

(P7, line 232-233)

 “First, this study randomized patients by the date of visit; however, the physicians were not blinded. Thus, there may be possibility of bias.”

Point 3: What were the factors/criteria for discharge?

Response 3: If the patient can tolerate pain and there were no complications associated with obstructive nephropathy, they were discharged based on the decision of the attending physician.

Point 4: Were urology consultants involved for any pts who were diagnosed with stones? If yes, in some or all pts? Involvement of a consultant would be a factor in LOS.

Response 4: An urology consultant was not involved in all the cases, only in cases of patients with obstructive nephropathy or uncontrolled pain. In our institution, urology consultant was available every day, every time. As you have indicated urology consultation may have increased ED LOS. However, this study is a randomized controlled study, and so it would not have a significant effect on the results.

Point 5: Lines 132-133, please correlate with group numbers (N) in Fig 1. It appears that 164 pts were enrolled and randomized, and 12 were removed due to group match erros, but where are the 28 pts who dropped out? If "dropped out" means excluded, this should not be mentioned as part of the enrolled sample. Please describe recruitment and enrollment procedures in more detail.

Response 5: We apologize for the error in reporting this. Twelve patients dropped out due to group mismatching. We have revised the manuscript per your advice, as follows:

(P4 lines 134–135)

“One hundred and sixty-four patients were enrolled, but 12 patients dropped out because of group mismatching”

Point 6: Did enrollment in both groups continue until each group reached exactly 76 patients? If so, did both groups reach the target at the same time? When and how were enrollment concluded?

Response 6: The enrollment did not finish at the same time. UG had two more weeks to enroll 76 patients. This study randomized the patients by the date of visit and patients suspected of renal colic did not come equally by date. Also, there were some months with 31 days.

As a result, this randomization procedure could introduce bias. We have described this as a limitation of the study, as follows:

(P7, line 232-233)

“There are several limitations. First, this study randomized by date of visit.”

Point 7: Table 1, please include comparative statistics between the groups.

Response 7: Based on your comment, we have revised Table 1.

Point 8: Authors should make clear that, based on their a priori exclusion of patients with complex history or with diagnoses requiring immediate intervention, the context of this trial is limited to clinically well patients with non-urgent renal colic and/or ureteral stones.

Response 8: As your comment, we excluded patients with fever, intra-abdominal malignancy, urogenital abnormality and interventional procedures. Consequently, as a result of these exclusion criteria, this study enrolled clinically well patients. We have described this as a limitation of our study in the manuscript, as follows:

(P8 line 242–243)

Seventh, as a result of exclusion criteria, we enrolled clinically well patients with non-urgent renal colic; thus results could not be generalized for clinically complicated patients.”

Point 9: Authors correctly point out that the timed saved in the LOS between the groups appears to have been from ED presentation to CT. It appears from the protocol that ED staff waited until results of all routine labs and UA were available to determine whether pts in the conventional group should then have CT. In routine practice, however, it would seem that the decision to pursue CT would be made in parallel rather than serially. Ultimately, if clinical suspicion (in addition to labs, UA, and/or POCUS) warrants cross-sectional imaging, it would be expected that ED staff would obtain CT, thus shortening the time to CT.

Response 9: I agree with your opinion. If the patient is clinically strongly suspected of renal colic, CT scans can be performed quickly. However, our institution is a tertiary university hospital, and there are many patients with severe diseases. It’s important to minimize fatal misdiagnosis and other complications. Therefore, in our institution we routinely carry out several laboratory tests and UA, which are then followed by CT scans. We designed this study based on the concept, “Detection of hydronephrosis by ultrasonography” to provide evidence for the usefulness of CT scans without awaiting other test results for diagnosing ureterolithiasis in patients with acute renal colic.

Point 10: Line 216, "are undergo CT scans" check grammar

Response 10: We apologize for the error. We have revised the manuscript, as follows:

(P7 lines 217–219)

It is important to minimize fatal misdiagnosis because our institution is the tertiary general hospital, and most patients suspected of having ureter stone due to high medical coverage in South Korea undergo CT scans”

Point 11: There is some evidence that clinically stable patients with high suspicion of stones on US may not need CT and can thus avoid radiation exposure. Do the findings of the present study have implications for US-only evaluation in the ED? 

Response 11: Although based on our current guidelines, the gold standard imaging tool for ureterolithiasis is abdomen non-contrast CT (PMID: 14960744, PMID: 26318710), CT scans have a radiation hazard. In line with your opinion, clinically stable patients with high suspicion of stones on US may not need CT.

However, hydronephrosis does not always mean that there is a ureter stone. Furthermore, in case of mild hydronephrosis, it can be misinterpreted from normal collecting duct by the physician. Moreover, CT scans can also help diagnose causes other than ureter stones. Therefore, we designed this study to provide evidence of the usefulness of CT scans for diagnosing ureterolithiasis in patients with acute renal colic.

Personally, I believe that in cases of patients who are stable and that have been risk stratified, the application of ultrasound alone is acceptable. However, the availability of this technique is dependent on the specific medical environment and countries and therefore, further research is needed to generalize its application.

Round 2

Reviewer 2 Report

the authors addressed all comments appropriately.

Reviewer 4 Report

Acceptable as revised